# Modulation of Diverse Procoagulant Venom Activities by Combinations of Platinoid Compounds

**DOI:** 10.3390/ijms22094612

**Published:** 2021-04-28

**Authors:** Vance G. Nielsen

**Affiliations:** Department of Anesthesiology, College of Medicine, University of Arizona, Tucson, AZ 85719, USA; vgnielsen333@gmail.com

**Keywords:** procoagulant venom, ruthenium, platinum, thrombelastography, carbon monoxide releasing molecule

## Abstract

Procoagulant snake venoms have been inhibited by the ruthenium containing compounds CORM-2 and RuCl_3_ separately, presumably by interacting with critical histidine or other sulfur-containing amino acids on key venom enzymes. However, combinations of these and other platinoid containing compounds could potentially increase, decrease or not affect the procoagulant enzyme function of venom. Thus, the purpose of this investigation was to determine if formulations of platinoid compounds could inhibit venom procoagulant activity and if the formulated compounds interacted to enhance inhibition. Using a human plasma coagulation kinetic model to assess venom activity, six diverse venoms were exposed to various combinations and concentrations of CORM-2, CORM-3, RuCl_3_ and carboplatin (a platinum containing compound), with changes in venom activity determined with thrombelastography. The combinations of CORM-2 or CORM-3 with RuCl_3_ were found to enhance inhibition significantly, but not in all venoms nor to the same extent. In sharp contrast, carboplatin-antagonized CORM-2 mediated the inhibition of venom activity. These preliminary results support the concept that platinoid compounds may inhibit venom enzymatic activity at the same or different molecular sites and may antagonize inhibition at the same or different sites. Further investigation is warranted to determine if platinoid formulations may serve as potential antivenoms.

## 1. Introduction

For the past few years this laboratory has focused on quantifying the effects of scores of snake venoms [1,2,3,4] and anticoagulant enzymes isolated from such venoms [5,6] on human plasmatic coagulation, with an emphasis on the inhibitory action of carbon monoxide (CO) on such anticoagulant activity. The source of site-directed CO application to these venoms in isolation prior to placement into human plasma was release from a ruthenium (Ru)-based carbon monoxide releasing molecule (tricarbonyldichlororuthenium(II) dimer (CORM-2)). The specificity of CO-mediated effects was by the concurrent exposure of venom to an inactive releasing molecule that had undergone a degradation [1,2,3,4,6] with the result that this molecule would not inhibit venom activity to the extent that CORM-2 had inhibited activity [1,2,3,4,6]. This paradigm is decades old, but it was challenged in the setting of K^+^ channel inhibition [7] and assessment of antibacterial activity [8] exerted by Ru(II) based carbon monoxide releasing molecules (CORM). These recent investigations determined that these Ru(II) CORM formed a transition state that, after releasing CO, would bind to histidine [7,8], methionine [8], glutathione [8], or cysteine [8]. In response to these new findings [7,8], it was subsequently reported that purified phospholipase A_2_ isolated from bee venom was inhibited by CORM-2 via a CO-independent mechanism, with inhibition of the venom by CORM-3 (tricarbonylchloro(glycinato)ruthenium) likely also dependent on a Ru(II)-based mechanism [9]. Thereafter, anticoagulant metalloproteinase activity in venoms collected from mambas was found to be inhibited by CORM-2 by a similar CO-independent mechanism [10], and, finally, the procoagulant activity exerted by metalloproteinases and serine proteases was found to be inhibited by CORM-2 and ruthenium(III) chloride (RuCl_3_) [11]. Thus, as suggested by the works of the past several months [9,10,11], it is the ruthenium species, not CO, that are binding to key anticoagulant/procoagulant venom enzymes in a heme-independent and perhaps irreversible fashion.

It is of interest that multiple Ru-based molecular species have been synthesized and investigated as potential chemotherapeutic agents to replace the toxic platinum-based medications (e.g., cisplatin, carboplatin) used to treat various cancers [12,13,14,15,16,17,18,19]. Thus, investigations have demonstrated that Ru(II)-based compounds covalently bind to histidine, methionine, glutathione, or cysteine [8,12,13,14], and Ru(III)-based compounds similarly bind histidine and cysteine [15,16,17,18,19]. The platinum (Pt)-based compounds cisplatin and carboplatin also bind histidine and methionine [20]. These binding characteristics of Ru compounds to specific amino acid residues may explain why CORM-2 and RuCl_3_ separately have inhibited snake venom and isolated enzyme activities [1,2,3,4,5,6,9,10,11], as highly conserved histidines and disulfide bridges that are critical to function are found in snake venom metalloproteinases (MP) [21,22], snake venom serine proteases (SP) [23,24] and phospholipase A_2_ (PLA_2_) [25,26]. Taken as a whole, small molecular weight Ru-based or Pt-based compounds may inhibit anticoagulant/procoagulant snake venom activity by binding to a hereto unappreciated Achilles’ heel of highly conserved amino acid residues essential to functions shared across multiple enzyme types.

However, the inhibitory effects of any class of compound are not just based on valance, but also on size, composition, and other characteristics that can change the affinity of a compound for a ligand. The structures of CORM-2, CORM-3, RuCl_3_ and carboplatin are displayed in Figure 1. It is of note that CORM-2 and RuCl_3_ have been found to have similar or no inhibitory effects on various procoagulant venoms when tested separately [11]. This finding opened the possibility that the Ru-based and Pt-based compounds may bind to the same critical amino acid residue, with perhaps different affinity to different residues that are enzymatically important, or perhaps to more than two molecular sites on any given enzyme. It should also be noted that the proteome of such venoms contains a great deal of similar or diverse enzymes with different effects on coagulation that summate into primarily anticoagulant or procoagulant activities [1,2,3,4]. To reiterate, these enzymes include SP, MP, kallikrein-like SP, and molecules that closely resemble human coagulation factors V (FV) and X (FX) [27,28,29,30,31,32,33,34,35], which are found in the indicated venoms displayed in Table 1. These venoms were chosen as they have already demonstrated marked vulnerability to inhibition by CORM-2 or RuCl_3_ in previous works [1,2,3,4,11,36]. Thus, they would serve well to achieve the subsequently presented goals.

Given the aforementioned molecular complexity, the goal of this manuscript was to determine the effects of CORM-2, CORM-3, carboplatin and RuCl_3_ exposure (separately and as a formulation) on a variety of diverse procoagulant snake venoms to provide insight into any interactions of the compounds on venom procoagulant activity as assessed by changes in human plasmatic coagulation monitored with thrombelastography.

## 2. Results

### 2.1. Assessment of CORM-2 and RuCl_3_, Separately and in Combination, on the Procoagulant Activity of B. moojeni, C. rhodostoma, E. leucogaster and O. microlepidotus

The subsequent results were obtained using concentrations, or fractions thereof, of the aforementioned venoms previously published using these methods [1,2,3,4,5,6,36]; specifically, *B. moojeni* venom had a final concentration of 2 µg/mL, *C. rhodostoma* a venom concentration of 5 µg/mL, *E. leucogaster* a venom concentration of 1 µg/mL and *O. microlepidotus* a venom concentration of 1 µg/mL in the plasma mixtures tested. Venom concentrations were selected on a performance basis wherein the activation of coagulation by the venom statistically exceeded the activation observed by contact activation with thrombelastographic cup and pin contact with plasma, as previously described [1,2,3,4,5,6,36]. All venom solutions were added as a 1% addition to the plasma mix in our thrombelastographic system [1,2,3,4,5,6,36]. This dilution is critical, as it reduces the concentration of CORM-2 to 1 µM, a concentration at which this compound does not affect coagulation kinetics [4]. Further, it has been demonstrated that concentrations of RuCl_3_ at or below 1 µM do not significantly affect human plasmatic coagulation, meaning that exposure of venom to up to 100 µM in isolation in this system should not affect the interpretation of changes in venom procoagulant activity [37]. With regard to the concentrations of CORM-2 and RuCl_3_ used in the isolated exposures, they were as follows: *B. moojeni* venom was exposed to 0–1 mM CORM-2 and 0–100 µM RuCl_3_; *C. rhodostoma* venom was exposed to 0–50 µM CORM-2 and 0–50 RuCl_3_; *E. leucogaster* venom was exposed to 0–100 µM CORM-2 and 0–100 µM RuCl_3_; and, *O. microlepidotus* venom to 0–100 µM CORM-2 and 0–100 µM RuCl_3_. Lastly, the thrombelastographic model utilized describes coagulation kinetics with the following three variables: time to maximum thrombus generation (TMRTG, minutes—a measure of time to onset of coagulation), maximum rate of thrombus generation (MRTG, dynes/cm^2^/sec—a measure of the velocity of clot growth) and total thrombus generation (TTG, dynes/cm^2^—a measure of clot strength). The results of exposing the four venoms to CORM-2 and RuCl_3_ separately or in combination are displayed in the following Figure 2 and Figure 3.

As seen in Figure 2’s left panels, exposure of *B. moojeni* venom to 1 mM CORM-2 (depicted as Ru(II)) resulted in a significant increase in TMRTG and decrease in MRTG values compared to CORM-2 naïve venom, indicative of inhibition of procoagulant activity. In contrast, exposure of *B. moojeni* venom to 100 µM RuCl_3_ (depicted as Ru(III)) did not significantly diminish procoagulant activity in this dataset. When *B. moojeni* venom was exposed to the combination of CORM-2 and RuCl_3_, TMRTG values were far more increased and MRTG values decreased compared to all other conditions, with the inhibition of the procoagulant activity due to significant interaction of the two Ru-based compounds. Lastly, there were no effects of the Ru-based compounds on the final clot strength generated by the procoagulant activity of *B. moojeni* venom except in the condition wherein both were present, resulting in TTG values significantly greater than the condition wherein venom alone was present.

With regard to the results obtained with *C. rhodostoma* venom, these are displayed in the right panels of Figure 2. Exposure of *C. rhodostoma* venom to 50 µM CORM-2 resulted in a significant increase in TMRTG and decrease in MRTG values compared to CORM-2 naïve venom, revealing procoagulant activity inhibition. Similarly, exposure of *C. rhodostoma* venom to 50 µM RuCl_3_ significantly diminished procoagulant activity, evidenced by an increase in TMRTG values and a decrease in MRTG values compared to venom not exposed to RuCl_3_. When *C. rhodostoma* venom was exposed to the combination of CORM-2 and RuCl_3_, TMRTG values were far more increased and MRTG values decreased compared to all other conditions, with the inhibition of the procoagulant activity assessed by changes in MRTG due to a significant interaction of the two Ru-based compounds. Lastly, there was a significant decrease in TTG values when the venom was exposed to both CORM-2 and RuCl_3_ compared to venom exposed to neither compound. In summary, these diverse venoms displayed an enhanced inhibition of procoagulant activity following exposure to the combination of CORM-2 and RuCl_3_ compared to separate exposures of either compound.

As for the next two venoms tested, the results obtained with *E. leucogaster* venom and *O. microlepidotus* venom are displayed in the left and right panels of Figure 3, respectively. With regard to *E. leucogaster* venom, exposure to 100 µM CORM-2 resulted in a significant increase in TMRTG and decrease in MRTG values compared to CORM-2 naïve venom, demonstrating procoagulant activity inhibition. Similarly, exposure of *E. leucogaster* venom to 100 µM RuCl_3_ significantly diminished procoagulant activity, evidenced by an increase in TMRTG values and a decrease in MRTG values compared to venom not exposed to RuCl_3_. However, the inhibition of procoagulant activity was significantly less than that observed with CORM-2. When *E. leucogaster* venom was exposed to the combination of CORM-2 and RuCl_3_, TMRTG values were significantly more increased compared to all other conditions. Inhibition of venom activity assessed by changes in TMRTG due to a significant interaction of the two Ru-based compounds was also present. Further, MRTG values were significantly decreased compared to venom without exposures and venom exposed to RuCl_3_ but not different from venom exposed only to CORM-2. Lastly, there were no significant changes in TTG values between the conditions.

The results obtained with *O. microlepidotus* venom are displayed in the right panels of Figure 3. Exposure of this venom to 100 µM CORM-2 resulted in a significant increase in TMRTG, decrease in MRTG, and decrease in TTG values compared to CORM-2 naïve venom, demonstrating procoagulant activity inhibition. In sharp contrast, exposure of *O. microlepidotus* venom to 100 µM RuCl_3_ did not significantly affect procoagulant activity. Lastly, when *O. microlepidotus* venom was exposed to the combination of CORM-2 and RuCl_3_, the decrease in procoagulant activity was not significantly different from venom exposed to CORM-2 alone but significantly more inhibited than venom without exposure to any compounds or exposed to RuCl_3_.

In conclusion, these series of experiments with these four diverse venoms demonstrated very different patterns of inhibition by CORM-2, RuCl_3_, or the combination of these two compounds.

### 2.2. Assessment of CORM-3 and RuCl_3_, Separately and in Combination, on the Procoagulant Activity of B. moojeni, C. rhodostoma, P. textilis and H. suspectum

For this third series of experiments, *B. moojeni* venom had a final plasma sample concentration of 2 µg/mL, *C. rhodostoma* a venom concentration of 5 µg/mL, *P. textilis* a venom concentration of 0.1 µg/mL, and *H. suspectum* venom a concentration of 10 µg/mL. All other aspects of the plasma mix used were similar to that of the previous series. The concentrations of CORM-2 and RuCl_3_ used in the isolated exposures were as follows: *B. moojeni* venom was exposed to 1 mM CORM-3 and 100 µM RuCl_3_; *C. rhodostoma* venom was exposed to 50 µM CORM-3 and 50 RuCl_3_; *P. textilis* venom was exposed to 100 µM CORM-2 and 100 µM RuCl_3_; and, *H. suspectum* venom to 1 mM CORM-2 and 100 µM RuCl_3_. The results of exposing the four venoms to CORM-3 and RuCl_3_ separately or in combination are displayed in the following Figure 4 and Figure 5.

As seen in Figure 4 in the left panels, exposure of *B. moojeni* venom to 1 mM CORM-3 or 100 µM RuCl_3_ resulted in a significant increase in TMRTG compared to additive naïve venom, indicative of inhibition of procoagulant activity. In contrast, exposure of *B. moojeni* venom to CORM-3 or RuCl_3_ did not significantly change MRTG values. Exposure to both Ru-based compounds significantly increased TMRTG values compared to all other conditions, and MRTG values were decreased compared to venom without exposure to additives or exposure to RuCl_3_. TTG values were significantly increased by either Ru-based compound but TTG values decreased to values observed with venom not exposed to additives. This change in TTG values resulted in a significant interaction between CORM-3 and RuCl_3_.

The results obtained with *C. rhodostoma* venom are displayed in the right panels of Figure 4. Exposure of this venom to 100 µM CORM-3 resulted in no significant effect on procoagulant activity. In sharp contrast, exposure of *C. rhodostoma* venom to 100 µM RuCl_3_ resulted in a significant increase in TMRTG and decrease in MRTG compared to RuCl_3_ naïve venom or CORM-3 exposed venom but not different from RuCl_3_ exposed venom. With regard to MRTG values, the combination of CORM-3 and RuCl_3_ resulted in values significantly smaller than RuCl_3_ naïve venom or CORM-3 exposed venom; however, MRTG values under these conditions were significantly greater than that associated with venom exposed to RuCl_3_ alone. Lastly, when *C. rhodostoma* venom was exposed to the combination of CORM-3 and RuCl_3_, TTG values were significantly smaller than those observed in the RuCl_3_ naïve venom or CORM-3 exposed venom sample.

Data obtained from experiments performed with *P. textilis* venom and *H. suspectum* venom are depicted in the left and right panels of Figure 5, respectively. P. textilis venom exposed to 100 µM CORM-3 or 100 µM RuCl_3_ resulted in a significant increase in TMRTG, decrease in MRTG, and increase in TTG values compared to additive naïve venom. Further, when exposed to both CORM-3 and RuCl_3_, TMRTG values significantly larger than and MRTG values significantly smaller than the other three conditions were observed. In contrast, TTG values observed after venom was exposed to both CORM-3 and RuCl_3_ were significantly smaller than in samples with venom exposed to either Ru-based compound separately. Lastly, CORM-3 and RuCl_3_ demonstrated significant interactions on venom activity, as seen in the two-way ANOVA analyses.

Data obtained from experiments performed with *H. suspectum* venom are presented in the right panel of Figure 5. Exposure of this venom to CORM-3 resulted in a significant increase in TMRTG values in plasma but no change in either MRTG or TTG values. Exposure of *H. suspectum* venom to RuCl_3_ resulted in no significant changes in any of the coagulation kinetic parameters. However, exposure to both CORM-3 and RuCl_3_ resulted in TMRTG values significantly greater than venom not exposed to additives or venom exposed to RuCl_3_. In contrast, MRTG values were significantly decreased by the combination of CORM-3 and RuCl_3_ compared to all other conditions. Not changes in TTG were noted between the conditions. In summary, the exposure of *H. suspectum* venom to CORM-3 and RuCl_3_ in various combinations resulted in significant but quantitatively small inhibition of procoagulant activity.

In conclusion, this series of experiments demonstrated a diverse response to CORM-3 mediated inhibition compared to CORM-2 attenuation of activity with four very different venoms.

### 2.3. Assessment of CORM-2 and Carboplatin, Separately and in Combination, on the Procoagulant Activity of B. moojeni and C. rhodostoma Venoms

As with the previous experiments, *B. moojeni* venom had a final concentration of 2 µg/mL and *C. rhodostoma* a venom concentration of 5 µg/mL. All other aspects of the plasma mix used are similar to that of the two previous series except that the venoms were exposed to different combinations of carboplatin and CORM-2. The concentrations of carboplatin and CORM-2 used in the isolated exposures were as follows: *B. moojeni* venom was exposed to 100 µM carboplatin (depicted as Pt(II)) and 1 mM CORM-2; and, *C. rhodostoma* venom was exposed to 100 µM carboplatin and 100 µM CORM-2. The results of exposing these two venoms to carboplatin and CORM-2 separately or in combination are displayed in the following Figure 6, with data generated with *B. moojeni* venom in the left panels and data obtained with *C. rhodostoma* venom in the right panels.

As seen in Figure 6 in the left panels, exposure of *B. moojeni* venom to carboplatin resulted in no significant change in any of the coagulation kinetic parameters compared to samples with venom without additives. CORM-2 exposure resulted in significantly increased TMRTG and decreased MRTG values compared to CORM-2 naïve or carboplatin exposed venom samples. When carboplatin and CORM-2 were combined, TMRTG values were significantly different from the other three conditions with the important observation that the addition of carboplatin to CORM-2 decreased TMRTG values compared to samples with CORM-2 exposure. The interaction between carboplatin and CORM-2 was significant for TMRTG values, with carboplatin opposing CORM-2 mediated inhibition of the procoagulant activity of this venom. Lastly, no statistically significant changes in TTG values were noted between the four conditions.

The results obtained with *C. rhodostoma* venom are displayed in the right panels of Figure 6. Exposure of this venom to 100 µM carboplatin resulted in no significant effect on procoagulant activity. In sharp contrast, exposure of *C. rhodostoma* venom to 100 µM CORM-2 resulted in a significant increase in TMRTG, decrease in MRTG and decrease in TTG values compared to CORM-2 naïve venom or carboplatin. Exposure of this venom to the combination of carboplatin and CORM-2 and CORM-2 resulted in TMRTG values significantly different from the values other three conditions with the important finding that the addition of carboplatin to CORM-2 decreased TMRTG values compared to samples with CORM-2 exposure. Aside from this singular difference in TMRTG values, there was no significant differences in MRTG and TTG values between venom exposed to CORM-2 alone or the combination of carboplatin and CORM-2. As with *B. moojeni* venom, the interaction between carboplatin and CORM-2 was significant for TMRTG values, with carboplatin opposing CORM-2 mediated inhibition of the procoagulant activity of *C. rhodostoma* venom.

In conclusion, carboplatin did not demonstrate any detectable effect on the procoagulant activity of these two venoms, but this compound did in some way partially block the inhibitory effect of CORM-2 on increasing TMRTG values, thus delaying the initiation of clot formation.

## 3. Discussion

The present investigation succeeded in capturing unique observations regarding the effects of four platinoid compounds with different valences on diverse procoagulant venom activities. As mentioned in the Introduction, it was entirely possible that any or all of the four compounds tested (CORM-2, CORM-3, RuCl_3_, carboplatin) might be expected to interact with similar molecular targets (e.g., histidine, methionine, disulfide bridges) within the enzymes of the venoms tested [8,12,13,14,15,16,17,18,19,20]. To be sure, the venoms contained metalloproteinases, serine proteases, kallikrein-like enzymes, Factor X-like enzymes, and/or Factor V-like activities [27,28,29,30,31,32,33,34,35], and the vipers and one lizard chosen evolved in geographically diverse areas of the world. This selection of compounds and venoms permitted remarkably different results to be documented that provide molecular insight into the complex interactions modifying procoagulant activity. For clarity, the various patterns of interaction of the platinoids utilized with venoms will be considered in the order of experimentation as subsequently presented.

With regard to the interaction of CORM-2 and RuCl_3_, *B. moojeni* inhibition was possibly secondary to different molecular sites on the procoagulant enzyme(s) inhibited. Specifically, a relatively silent inhibitory interaction by RuCl_3_ only became important when CORM-2 was introduced. As for *C. rhodostoma* venom, it appeared that both Ru-based compounds individually inhibited procoagulant activity to an equivalent extent, and when combined, significantly inhibited activity more than when only one inhibitor was present. In the case of *E. leucogaster* venom procoagulant activity, CORM-2 was a more significant inhibitor than RuCl_3_, but when combined, inhibition was somewhat greater than when the venom was exposed to CORM-2 alone. Lastly, *O. microlepidotus* venom procoagulant activity was only inhibited by CORM-2, with the presence of RuCl_3_ not making any difference in activity without or with CORM-2 presence. Considered as a whole, the results point to potential diversity in binding sites by CORM-2 and RuCl_3_, associated with an equally unpredictable inhibitory effect.

The experiments involving venom exposures to CORM-3 and RuCl_3_ also revealed diverse patterns of procoagulant activity inhibition. In the case of *B. moojeni* venom, insignificant inhibition by both CORM-3 and RuCl_3_ were noted (Figure 4), but the inhibitory effects were far more diminutive than that observed with CORM-2 (Figure 1). The reason that the procoagulant activity of venom after RuCl_3_ exposure was significantly different from venom not exposed to this compound in Figure 4 but not in Figure 1 is a statistical phenomenon—the mean values and variance of the other conditions in Figure 1 overshadowed the condition of RuCl_3_-exposed venom, but not so in Figure 4. Nevertheless, the effects of RuCl_3_ on this venom were quantitatively very small. As for *C. rhodostoma* venom exposed to CORM-3 and RuCl_3_, CORM-3 had no discernable effect on procoagulant activity, and when combined with RuCl_3_, it appeared that CORM-3 partially antagonized RuCl3-mediated inhibition of procoagulant activity based on an increase in MRTG values compared to venom samples exposed to RuCl_3_ alone (Figure 4). With regard to *P. textilis* venom procoagulant activity, CORM-3 and RuCl_3_ had equivalent inhibitory effects, with the combination of the two Ru-based compounds significantly interacting and significantly increasing inhibition of activity. Finally, as for *H. suspectum* venom procoagulant activity, only the combination of CORM-3 and RuCl_3_ exerted meaningful inhibition of activity. In summary, unlike CORM-2, CORM-3 was unpredictably far less potent as a direct inhibitor of procoagulant activity in some cases, and unpredictably enhanced or partially inhibited RuCl_3_-mediated inhibition of procoagulant activity.

Experimentation involving carboplatin and CORM-2 was in some ways the most fascinating. Simply put, carboplatin by itself had no detectable effects on the procoagulant activity of *B. moojeni* and *C. rhodostoma* venom; however, carboplatin exposure significantly antagonized CORM-2 mediated inhibition of venom procoagulant activity as evidenced by decreased TMRTG values (Figure 6). While at face value it seems simple enough to imagine a competition between carboplatin and CORM-2 on a common molecular site of similar enzymes, it is far more difficult to explain with such a paradigm why MRTG values did not change as well. As a rule, increases or decreases in thrombin generation are accompanied by concordant decreases or increases in TMRTG values and increases or decreases in MRTG values, respectively. Taken as a whole, while it is clear that a Pt-based compound appears to partially antagonize a Ru-based compound mediated inhibition of procoagulant activity, the precise molecular explanation for the coagulation kinetic changes observed remains to be elucidated.

It is important to note that even if combinations of CORM-2, CORM-3 and RuCl_3_ are demonstrated in vitro to be more effective than either separately in inhibiting venom enzymatic activity, that is a far cry from being able to be assured that this formulation is not toxic in vivo. To be sure, systemic/local CORM-2 or CORM-3 administrations that would result in circulating or regional concentrations as great or greater than that reported here in vivo has been demonstrated to be benign in animal models too numerous to fully cite, including rabbits [38,39] and rats [40,41]. It is also important to note that the concentrations of CORM-2, CORM-3 and RuCl_3_ are small, and are dissolved in PBS at pH 7.4—a not particularly toxic solution. With regard to the nature of each Ru compound, it is likely that RuCl_3_ is forming some sort of phosphate ion following the loss of a Cl^-^ in solution that may not be a high energy species, given the physiological milieu. RuCl_3_ is only hemostatically active in human plasma if dissolved in PBS and not active if suspended in water—but only at 100 times the concentration in plasma as used here in experimentation [37]. In contrast, CORM-2 (and probably CORM-3) likely form a transient, high energy radical during carbon monoxide release for a couple of reasons. First, CORM-2 synthesis involves exposing RuCl_3_ to high, nonphysiological temperatures in the presence of various solvents containing carbon monoxide-donating compounds or in the presence of carbon monoxide gas at several atmospheres of pressure for hours (patents.google.com/patent/CN107033191A/en; accessed on 10 March 2021). Second, CORM-2 that is inactivated (presumably by complete release of carbon monoxide) does not have the same carbon monoxide-independent inhibitory effects on various enzymes and ion channels as does the form of the compound actively releasing carbon monoxide [7,8,9,10,11]. Taken as a whole, more investigation is needed to determine the suitability of a formulation of such Ru based compounds, which contain molecules that modulate enzymatic function based not just on valance but also on ionic/radical states.

The use of venom rather than isolated enzymes is a limitation of the present study, but the utilization of thrombelastography to assess venom-mediated procoagulant activity in human plasma is very nearly an exercise in individual enzyme interrogation. The initiation of coagulation is a complex threshold event that rapidly consumes available substrate (e.g., fibrinogen). In the case of enzymes that activate prothrombin directly or other proximate serine proteases of the coagulation system, one particular enzyme will be in sufficient abundance with superior rate of catalysis to outcompete other venom enzymes that are present to initiate coagulation—making this one enzyme the *de facto* “procoagulant activity” responsible for the coagulation kinetics recorded via thrombelastography. In the case of enzymes that are “thrombin-like”, as in the example of the serine protease ancrod found in the venom of *C. rhodostoma* [28], the enzyme will catalyze fibrinogen and factor XIII far more quickly than the contact protein system engaged by the plastic cup and pin, so that the substrates will be consumed by the venom enzyme before the endogenous plasma serine proteases are generated. Or in the case of *H. suspectum* venom, the hemostatically active enzyme is a kallikrein-like enzyme that rapidly initiates contact protein activation of coagulation before factor XII is able to do so as recently reviewed [36]. Given the aforementioned, subsequent inhibition of venom mediated procoagulant activity by the various Ru-based compounds can reasonably be assumed to be primarily due to inactivation of the very same enzymes and likely any other similar enzymes present that had previously been outcompeted by the predominant species. In conclusion, despite being composed of several enzymes that might engage the process of coagulation, thrombelastography permits assessment of the kinetically most important enzyme responsible for whole venom procoagulant activity, making the technique an exercise in molecular analysis.

Another important matter to consider when interpreting the data is found in the background literature generated by this laboratory that guided the selection of venom concentrations and inhibitor concentrations [1,2,3,4,5,6,36]. The thrombelastographic paradigm used to characterize venom activity as procoagulant or anticoagulant is performance-based, meaning that venom is added to plasma at a concentration that results in coagulation kinetics that are markedly and statistically significantly different from results observed in normal plasma [1,2,3,4,5,6,36]. For example, for a venom concentration to be selected for experimentation, it would have to be sufficient to at least half the time to onset of coagulation (e.g., TMRTG) and/or double the maximum velocity of clot growth (e.g., MRTG) [1,2,3]. This would assure that endogenous processes that would initiate thrombin generation and clot formation via contact protein system engagement are outcompeted kinetically, allowing assessment of the venom enzymes studied. This is the justification for the concentrations of venoms used in the present study. It is also of interest that the concentrations of CORM-2, CORM-3 and RuCl_3_ were chosen based on the observations obtained in the aforementioned articles [1,2,3,4,5,6,36] wherein the goal was to determine what concentration would result in a statistically significant reduction in either venom anticoagulant or procoagulant activity. After assessing sixty different venoms over the past five years, it appears that standardized concentrations of CORM-2 (100 µM for most venoms, 1 mM for resistant venoms) have allowed investigation of the efficacy of inhibition by this class of molecule. Using this standard, the present investigation is the first to compare, on a mole-to-mole basis, the efficacy and interactions of different Ru-based molecules and one Pt-based molecule on potential inhibition of the diverse venoms investigated. The variations in concentration and combination of platinoid compound used was designed not to obtain an idealized determination of optimal inhibitory concentrations, but instead to determine in an exploratory spirit if there was variation in inhibitory response as a basis to postulate that multiple molecular sites of interaction existed within the procoagulant enzymes involved. The data would indicate that this goal was achieved, and that his data should serve as a preliminary, hypothesis-generating exercise into these remarkably complex venom-mediated effects on coagulation.

In conclusion, the present study demonstrated that hereto unappreciated binding sites on procoagulant enzymes within diverse venoms with complex proteomes may be vulnerable to inhibition of activity by a variety of Ru-based compounds with different valences, separately or as a formulation. Further, a Pt-based compound was found to antagonize Ru-based compound mediated inhibition of the procoagulant activity of diverse venoms. These observations provide molecular insight into the potentially multiple sites present on such procoagulant enzymes that may be therapeutic targets when designing small molecular weight antivenom molecules. Future investigation is justified to determine the differential response of hemostatically active venoms (e.g., procoagulant, anticoagulant, neurotoxic) to Ru-based compounds of multiple valences and molecular size, in isolation and in formulations of two or more compounds.

## 4. Materials and Methods

### 4.1. Chemicals and Human Plasma

Calcium-free phosphate buffered saline (PBS), CORM-2, CORM-3, ruthenium chloride and carboplatin were obtained from Millipore Sigma (Saint Louis, MO, USA). Venoms dissolved in PBS (50 mg/mL) were obtained from archived, never thawed aliquots maintained at −80 °C in the laboratory that were used in previous investigations [1,2,3,4,11,36]. *Bothrops moojeni* and *Calloselasma rhodostoma* venoms were obtained originally from the National Natural Toxins Research Center at Texas A&M University (Kingsville, TX, USA). Additionally, *Echis leucogaster*, *Heloderma suspectum*, *Oxyuranus microlepidotus* and *Pseudonaja textilis* venoms were originally purchased from Mtoxins (Oshkosh, WI, USA). Calcium chloride (200 mM) was obtained from Haemonetics Inc., Braintree, MA, USA. Pooled normal human plasma (George King Bio-Medical, Overland Park, KS, USA) that was sodium citrate anticoagulated and maintained at −80 °C was used.

### 4.2. Thrombelastographic Analyses

The volumes of subsequently described plasmatic and other additives summed to a final volume of 360 µL. Samples were composed of 320 µL of plasma; 16.4 µL of PBS, 20 µL of 200 mM CaCl_2_, and 3.6 µL of PBS or venom mixture, which were pipetted into a disposable cup in a thrombelastograph^®^ hemostasis system (Model 5000, Haemonetics Inc., Braintree, MA, USA) at 37 °C, and then rapidly mixed by moving the cup up against and then away from the plastic pin five times. The following viscoelastic parameters described previously [6,7,10,12,13] were measured: time to maximum rate of thrombus generation (TMRTG): this is the time interval (minutes) observed prior to maximum speed of clot growth; maximum rate of thrombus generation (MRTG): this is the maximum velocity of clot growth observed (dynes/cm^2^/second); and total thrombus generation (TTG, dynes/cm^2^), the final viscoelastic resistance observed after clot formation. Data were collected until a stable maximum amplitude was observed with minimal change for 3 min as determined by the software.

### 4.3. Exposures of Venoms to CORM-2, CORM-3, RuCl_3_ and Carboplatin

A selection of venoms was exposed to CORM-2 concentrations (or fractions thereof) demonstrated to inhibit procoagulant activity and placed into plasma at the final venom concentrations previously used in this plasma based, thrombelastographic system [1,2,3,4]. Indicated venoms were also exposed to CORM-3, RuCl_3_ and carboplatin in various combinations subsequently presented. The specific exposures for each venom are as follows.

*B. moojeni*. This venom was exposed to 0 or 1 mM CORM-2 or CORM-3 in the presence of 0 or 100 µM RuCl_3_ in PBS for at least 5 min at room temperature prior to no additives, exposed to 1 mM CORM-2 or CORM-3, exposed to 100 µM RuCl_3_ in PBS, or exposed to 1 mM CORM-2 or CORM-3 and 100 µM RuCl_3_ in PBS for at least 5 min at room temperature prior to placement into plasma followed immediately with commencement of thrombelastographic analysis. This venom was also exposed to no additives, 1 mM CORM-2, 100 µM carboplatin, or exposed to 1 mM CORM-2 and 100 µM carboplatin in another series of experiments. The final concentration of this venom in plasma was 2 µg/mL.

*C. rhodostoma*. This venom was exposed to no additives, exposed to 50 µM CORM-2 or CORM-3, exposed to 50 µM RuCl_3_ in PBS, or exposed to 50 µM CORM-2 or CORM-3 and 50 µM RuCl_3_ in PBS for at least 5 min at room temperature prior to placement into plasma followed immediately with commencement of thrombelastographic analysis. 0 or 50 µM CORM-2 or CORM-3 in the presence of 0 or 50 µM RuCl3 in PBS for at least 5 min at room temperature prior to placement into plasma followed immediately with commencement of thrombelastographic analysis. This venom was also exposed to no additives, 100 µM CORM-2, 100 µM carboplatin, or 100 µM CORM-2 and 100 µM carboplatin in another series of experiments. The final concentration of this venom in plasma was 5 µg/mL.

*E. leucogaster*. This venom was exposed no additives, 100 µM CORM-2, 100 µM RuCl_3_, or 100 µM CORM-2 and 100 µM RuCl_3_ in PBS for at least 5 min at room temperature prior to placement into plasma, followed immediately with commencement of thrombelastographic analysis. The final concentration of this venom in plasma was 1 µg/mL.

*O. microlepidotus*. This venom was exposed to no additives, 100 µM CORM-2, 100 µM RuCl_3_, or 100 µM CORM-2 and 100 µM RuCl_3_ in PBS for at least 5 min at room temperature prior to placement into plasma, followed immediately with commencement of thrombelastographic analysis. The final concentration of this venom in plasma was 1 µg/mL.

*P. textilis*. This venom was exposed to to no additives, 100 µM CORM-3, 100 µM RuCl_3_, or 100 µM CORM-3 and 100 µM RuCl_3_ in PBS for at least 5 min at room temperature prior to placement into plasma, followed immediately with commencement of thrombelastographic analysis. The final concentration of this venom in plasma was 0.1 µg/mL.

*H. suspectum*. This venom was exposed to no additives, 1 mM CORM-3, 100 µM RuCl_3_, or 1 mM CORM-3 and 100 µM RuCl_3_ in PBS for at least 5 min at room temperature prior to placement into plasma, followed immediately with commencement of thrombelastographic analysis. The final concentration of this venom in plasma was 10 µg/mL.

Given the aforementioned, the experimental conditions utilized were: (1) V condition—venom in PBS; (2) Ru(II) condition—venom exposed to CORM-2 or CORM-3; (3) Ru(III) condition—venom exposed to RuCl_3_; (4) Ru(II + III) condition—venom exposed to CORM-2 or CORM-3 and RuCl_3_ simultaneously; (5) Pt(II) condition—venom exposed to carboplatin; and (6) Pt + Ru condition—venom exposed to carboplatin and CORM-2. After the 5 min period at room temperature, 3.6 µL of one of these solutions was added to the plasma sample in the plastic thrombelastograph cup.

### 4.4. Statistical Analyses and Graphics

Data are presented as mean ± SD. Graphics were generated with a commercially available program (Origin2020b, OriginLab Corporation, Northampton, MA, USA). Experimental conditions were composed of n = 6 replicates per condition as this provides a statistical power > 0.8 with *p* < 0.05 utilizing these techniques [1,2,3,4]. A statistical program was used for one-way analyses of variance (ANOVA) comparisons between conditions, followed by Holm–Sidak post hoc analysis. Additional analysis with two-way ANOVA was performed to detect significant interactions between CORM-2 and RuCl_3_, CORM-3 and RuCl_3_, and CORM-2 and carboplatin regarding changes in venom procoagulant activity. All analyses were performed with commercial software (SigmaPlot 14, Systat Software, Inc., San Jose, CA, USA). *p* < 0.05 was considered significant.

## Figures and Tables

**Figure 1 ijms-22-04612-f001:**
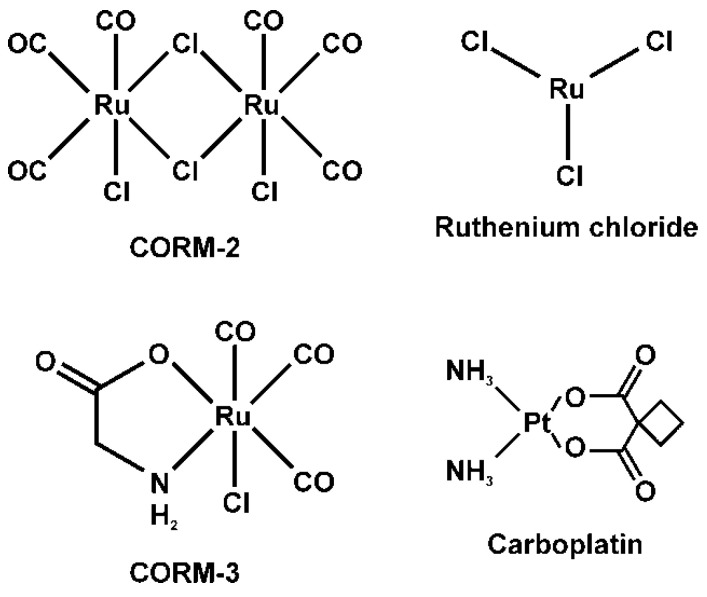
The structures and names of the Ru-based and platinum (Pt)-based compounds used in the present study.

**Figure 2 ijms-22-04612-f002:**
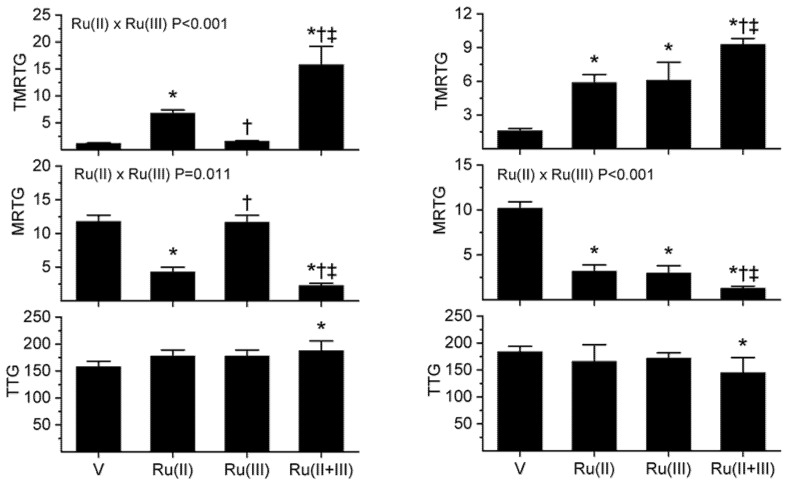
Procoagulant activity of *B. moojeni* venom (left panels) and *C. rhodostoma* venom (right panels) in plasma after exposure to CORM-2 (Ru(II)), RuCl_3_ (Ru(III)) or both (Ru(II + III)) in isolation. Data is presented as mean ± SD. V = venom; Ru(II) = V + CORM-2 in PBS; Ru(III) = V + RuCl_3_; Ru(II + III) = V + CORM-2 and RuCl_3_. * *p* < 0.05 vs. V; † *p* < 0.05 vs. Ru(II); ‡ *p* < 0.05 vs. Ru(III) via one-way analysis of variance (ANOVA) with Holm–Sidak post hoc test. Significant interactions between CORM-2 and RuCl_3_ (Ru(II) × Ru(III)) determined with two-way ANOVA are displayed within individual parameter graphics.

**Figure 3 ijms-22-04612-f003:**
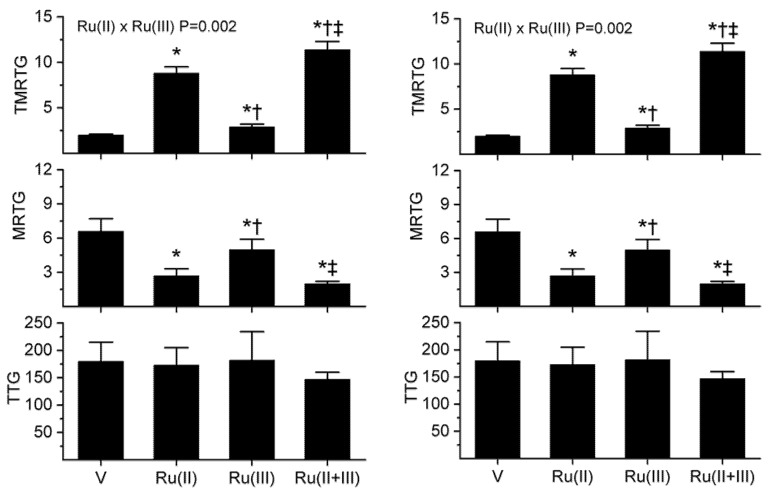
Procoagulant activity of *E. leucogaster* venom (left panels) and *O. microlepidotus* venom (right panels) in plasma after exposure to CORM-2 (Ru(II)), RuCl_3_ (Ru(III)) or both (Ru(II + III)) in isolation. Data is presented as mean ± SD. V = venom; Ru(II) = V + CORM-2 in PBS; Ru(III) = V + RuCl_3_; Ru(II + III) = V + CORM-2 and RuCl_3_. * *p* < 0.05 vs. V; † *p* < 0.05 vs. Ru(II); ‡ *p* < 0.05 vs. Ru(III). Significant interactions between CORM-2 and RuCl_3_ (Ru(II) × Ru(III)) determined with two-way ANOVA are displayed within individual parameter graphics.

**Figure 4 ijms-22-04612-f004:**
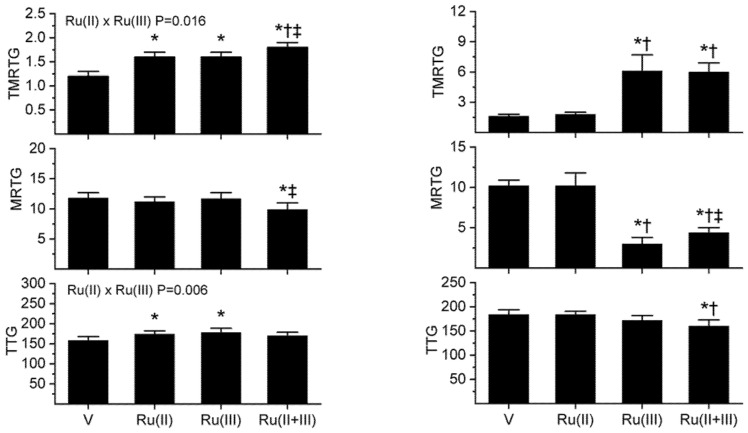
Procoagulant activity of *B. moojeni* venom (left panels) and *C. rhodostoma* venom (right panels) in plasma after exposure to CORM-3 (Ru(II)), RuCl_3_ (Ru(III)) or both (Ru(II + III)) in isolation. Data is presented as mean ± SD. V = venom; Ru(II) = V + CORM-3 in PBS; Ru(III) = V + RuCl_3_; Ru(II + III) = V + CORM-3 and RuCl_3_. * *p* < 0.05 vs. V; † *p* < 0.05 vs. Ru(II); ‡ *p* < 0.05 vs. Ru(III). Significant interactions between CORM-3 and RuCl_3_ (Ru(II) × Ru(III)) determined with two-way ANOVA are displayed within individual parameter graphics.

**Figure 5 ijms-22-04612-f005:**
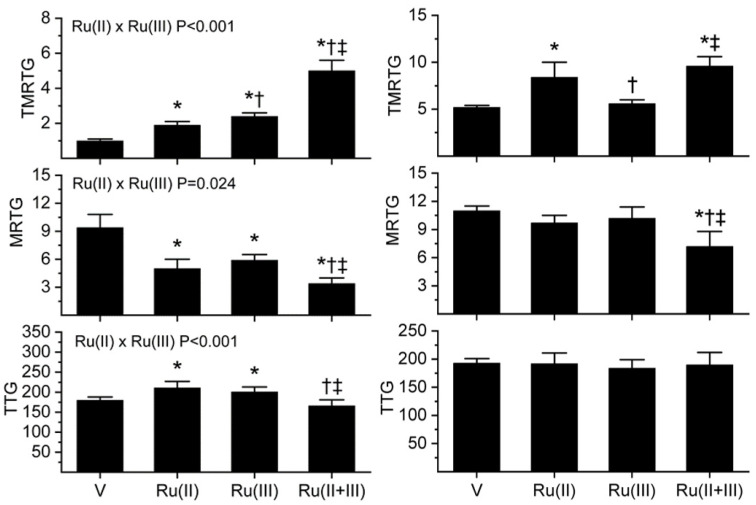
Procoagulant activity of *P. textilis* venom (left panels) and *H. suspectum* venom (right panels) in plasma after exposure to CORM-3 (Ru(II)), RuCl_3_ (Ru(III)) or both (Ru(II + III)) in isolation. Data is presented as mean ± SD. V = venom; Ru(II) = V + CORM-3 in PBS; Ru(III) = V + RuCl_3_; Ru(II + III) = V + CORM-3 and RuCl_3_. * *p* < 0.05 vs. V; † *p* < 0.05 vs. Ru(II); ‡ *p* < 0.05 vs. Ru(III). Significant interactions between CORM-3 and RuCl_3_ (Ru(II) × Ru(III)) determined with two-way ANOVA are displayed within individual parameter graphics.

**Figure 6 ijms-22-04612-f006:**
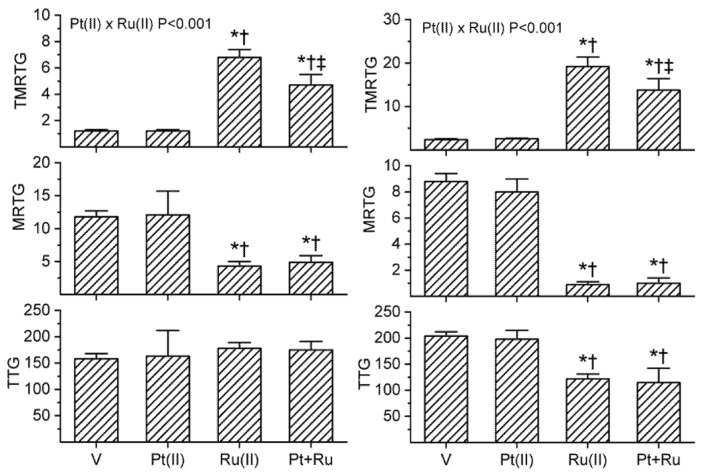
Procoagulant activity of *B. moojeni* venom (left panels) and *C. rhodostoma* venom (right panels) in plasma after exposure to carboplatin (Pt(II)), CORM-2 (Ru(II)), or both (Pt + Ru) in isolation. Data is presented as mean ± SD. V = venom; Pt(II) = V + carboplatin in PBS; Ru(II) = V + CORM-2; Pt + Ru = V + carboplatin and CORM-2. * *p* < 0.05 vs. V; † *p* < 0.05 vs. Pt(II); ‡ *p* < 0.05 vs. Ru(II) via one-way analysis of variance (ANOVA) with Holm–Sidak post hoc test. Significant interactions between carboplatin and CORM-2 (Pt(II) × Ru(II)) determined with two-way ANOVA are displayed within individual parameter graphics.

**Table 1 ijms-22-04612-t001:** Properties of procoagulant snake venoms investigated.

Species	Common Name	Proteome
*Bothrops moojeni* [27]	Brazilian Lancehead	SP, MP
*Calloselasma rhodostoma* [28]	Malayan Pit Viper	SP, MP
*Echis leucogaster* [29,30,31]	White-Bellied Carpet Viper	SP, MP
*Heloderma suspectum* [32,33]	Gila Monster	Kallikrein-like SP
*Oxyuranus microlepidotus* [34]	Inland Taipan	Factor V-like, SP, MP
*Pseudonaja textilis* [35]	Eastern Brown Snake	Factor V, X-like SP, MP

## Data Availability

All data generated are presented in this manuscript.

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
