# Peer review of "Modulation of Diverse Procoagulant Venom Activities by Combinations of Platinoid Compounds"

_ijms, 2021, doi:10.3390/ijms22094612_

Round 1
Reviewer 1 Report
The manuscript has investigated in detail the modulation of diverse procoagulant venom activities with Ruthenium-based and/or Platinoid-based compounds. The author measured the changes of procoagulant parameters using human plasmatic coagulation system in the absence or the presence, sometimes in combination of the testing chemicals.
Minor,
Numerous typing errors, some of which are already marked in the manuscript review attached, should be corrected before acceptance.
The concentrations of testing chemicals seem to be very important for the present study. It is not clear how the concentrations of each chemical have been picked up in the present study to show the interactions. Even though there might be some literature about these, optimum condition (concentration) result can be different depending on person, minor modifications, or something else. Furthermore, the interplay between two or more elements can be critically affected whether they were at the optimum concentration or sub-optimum concentration. Therefore, it is always trusty only when it is tested in the laboratory in the preliminary study. This critical issue should be adequately addressed in the manuscript.
Major,
Although it is an interesting subject and might be potentially important, the main results seem to be more phenomenal and not much of mechanistic. It would be better if there are some more evidence or proof for the conclusion. Animal venoms are usually composed of multiple components, which give a high degree of complexity in interpreting their data. For example, an animal venom can have metalloproteinase-like activities, in which there can be many variants and subtypes in a single bit. Therefore, the interpretations of data need to be careful unless otherwise there are some solid evidence.

Author Response
Reviewer #1
“The manuscript has investigated in detail the modulation of diverse procoagulant venom activities with Ruthenium-based and/or Platinoid-based compounds. The author measured the changes of procoagulant parameters using human plasmatic coagulation system in the absence or the presence, sometimes in combination of the testing chemicals.”
“Minor,”
“Numerous typing errors, some of which are already marked in the manuscript review attached, should be corrected before acceptance.”
Thank you for spending the time to do this for me. I have corrected the errors the reviewer was kind enough to identify.
“The concentrations of testing chemicals seem to be very important for the present study. It is not clear how the concentrations of each chemical have been picked up in the present study to show the interactions. Even though there might be some literature about these, optimum condition (concentration) result can be different depending on person, minor modifications, or something else.”
I appreciate this comment and have added more detail concerning the concentrations of venom and inhibitor. This is now outlined in a new paragraph in the Discussion.
“Furthermore, the interplay between two or more elements can be critically affected whether they were at the optimum concentration or sub-optimum concentration. Therefore, it is always trusty only when it is tested in the laboratory in the preliminary study. This critical issue should be adequately addressed in the manuscript.”
Again, this is similar to the issue mentioned in the previous comment by the reviewer. I include the new paragraph subsequently, and I hope that it satisfies the concern of the reviewer:
“Another important matter to consider when interpreting the data is found in the background literature generated by this laboratory that guided the selection of venom concentrations and inhibitor concentrations [1-6,36]. The thrombelastographic paradigm used to characterize venom activity as procoagulant or anticoagulant is performance-based, meaning that venom is added to plasma at a concentration that results in coagulation kinetics that are markedly and statistically significantly different from results observed in normal plasma [1-6,36]. For example, for a venom concentration to be selected for experimentation, it would have to be sufficient to at least half the time to onset of coagulation (e.g., TMRTG) and/or double the maximum velocity of clot growth (e.g., MRTG) [1-3]. This would assure that endogenous processes that would initiate thrombin generation and clot formation via contact protein system engagement are outcompeted kinetically, allowing assessment of the venom enzymes studied. This is the justification for the concentrations of venoms used in the present study. Also of interest, the concentrations of CORM-2, CORM-3 and RuCl3 were chosen based on the observations obtained in the aforementioned articles [1-6,36] wherein the goal was to determine what concentration would result in a statistically significant reduction in either venom anticoagulant or pro-coagulant activity. After assessing sixty different venoms over the past five years, it appears that standardized concentrations of CORM-2 (100 µM for most venoms, 1 mM for resistant venoms) have allowed investigation of the efficacy of inhibition by this class of molecule. Using this standard, the present investigation is the first to compare, on a mole-to-mole basis, the efficacy and interactions of different Ru-based molecules and one Pt-based molecule on potential inhibition of the diverse venoms investigated. The variations in concentration and combination of platinoid compound used was designed not to obtain an idealized determination of optimal inhibitory concentrations, but instead to determine in an exploratory spirit if there was variation in inhibitory response as a basis to postulate that multiple molecular sites of interaction existed within the procoagulant enzymes involved. The data would indicate that this goal was achieved, and that his data should serve as a preliminary, hypothesis-generating exercise into these remarkably com-plex venom mediated effects on coagulation.”
“Major,”
“Although it is an interesting subject and might be potentially important, the main results seem to be more phenomenal and not much of mechanistic. It would be better if there are some more evidence or proof for the conclusion. Animal venoms are usually composed of multiple components, which give a high degree of complexity in interpreting their data. For example, an animal venom can have metalloproteinase-like activities, in which there can be many variants and subtypes in a single bit. Therefore, the interpretations of data need to be careful unless otherwise there are some solid evidence.”
Thank you for this important question. I have now added a detailed paragraph that outlines how using thrombelastography allows molecular insight into the source of the procoagulant activity displayed by any given venom. This paragraph is as follows:
“The use of venom rather than isolated enzymes is a limitation of the present study, but the utilization of thrombelastography to assess venom mediated procoagulant activity in human plasma is very nearly an exercise in individual enzyme interrogation. The initiation of coagulation is a complex threshold event that rapidly consumes available substrate (e.g., fibrinogen). In the case of enzymes that activate prothrombin directly or other proximate serine proteases of the coagulation system, one particular enzyme will be in sufficient abundance with superior rate of catalysis to outcompete other venom enzymes that are present to initiate coagulation – making this one enzyme the de facto “procoagulant activity” responsible for the coagulation kinetics recorded via thrombelastography. In the case of enzymes that are “thrombin-like”, as in the example of the serine protease ancrod found in the venom of C. rhodostoma [28], the enzyme will catalyze fibrinogen and factor XIII far more quickly than the contact protein system engaged by the plastic cup and pin, so that the substrates will be consumed by the venom enzyme before the endogenous plasma serine proteases are generated. Or in the case of H. suspectum venom, the hemostatically active enzyme is a kallikrein-like enzyme that rapidly initiates contact protein activation of coagulation before factor XII is able to do so as recently reviewed [36]. Given the aforementioned, subsequent inhibition of venom mediated procoagulant activity by the various Ru-based compounds can reasonably be assumed to be primarily due to inactivation of the very same enzymes and likely any other similar enzymes present that had previously been outcompeted by the predominant species. In conclusion, despite being composed of several enzymes that might engage the process of coagulation, thrombelastography permits assessment of the kinetically most important enzyme responsible for whole venom procoagulant activity, making the technique an exercise in molecular analysis.”
It is my hope that the reviewer now appreciates that thrombelastography allows a molecular assessment as one enzyme will “get there first” to initiate thrombin generation or by polymerizing fibrinogen before endogenous thrombin can be formed by contact protein activation.
Reviewer 2 Report
The authors have already announced at IJMS last year that the ruthenium compound has proved the neutralizing effect of snake venom on the blood coagulation system by in vitro experiments using a combination of several ruthenium compounds.
- Pointing out the whole; We have observed the effects of adding platinum compound anticancer agents to the combination of ruthenium compounds, but the reviewers cannot understand the significance of conducting this combination experiment. I think that the reader can read this paper without any discomfort by explaining the significance of combining one type of platinum compound with the ruthenium compound in the introduction and discussion in a little more detail.
Author Response
Reviewer #2
“The authors have already announced at IJMS last year that the ruthenium compound has proved the neutralizing effect of snake venom on the blood coagulation system by in vitro experiments using a combination of several ruthenium compounds.”
Thank you for this comment, but I must point out that this viewpoint is not supported by the publication the reviewer cites. The article in question is “Nielsen, V.G. Ruthenium, Not Carbon Monoxide, Inhibits the Procoagulant Activity of Atheris, Echis, and Pseudonaja Venoms. Int J Mol Sci 2020, 21, 2970.” I did not use a combination of two ruthenium compounds, and only two compounds (CORM-2 and ruthenium chloride) were separately tested. The present work is the first to test combinations of several compounds (CORM-2, CORM-3, ruthenium chloride, carboplatin). The rationale for combining these ruthenium and platinum containing compounds is found in the Introduction of the present manuscript in review.
“Pointing out the whole; We have observed the effects of adding platinum compound anticancer agents to the combination of ruthenium compounds, but the reviewers cannot understand the significance of conducting this combination experiment.”
I appreciate this comment, and the significance of conducting these series of experiments is found in the Introduction. I provided a rationale for testing Pt based compounds based on their ability to bind to histidine residues critical to snake venom enzyme function. I have added just a few words to make this clearer to the readership. Specifically, as can be found on page 2, the following text provides the rationale for testing various compounds:
“Of interest, multiple Ru-based molecular species have been synthesized and investigated as potential chemotherapeutic agents to replace the toxic platinum-based medications (e.g., cisplatin, carboplatin) to treat various cancers [12-19]. Thus, investigations have demonstrated that Ru(II) based compounds covalently bind to histidine, methionine, glutathione, or cysteine [8,12-14], and Ru(III) based compounds similarly bind histidine and cysteine [15-19]. The platinum(Pt)-based compounds cisplatin and carboplatin also bind histamine and methionine [20]. These binding characteristics of Ru compounds to specific amino acid residues may explain why CORM-2 and RuCl3 separately have inhibited snake venom and isolated enzyme activities [1-6,9-11] as highly conserved histidines and disulfide bridges that are critical to function are found in snake venom metalloproteinases (MP) [21,22], snake venom serine proteases (SP) [23,24] and phospholipase A2 (PLA2) [25,26]. Taken as a whole, small molecular weight, Ru-based or Pt-based compounds may inhibit anticoagulant/procoagulant snake venom activity by binding to a hereto unappreciated Achilles’ heel of highly conserved amino acid residues essential to function shared across multiple enzyme types.”
“I think that the reader can read this paper without any discomfort by explaining the significance of combining one type of platinum compound with the ruthenium compound in the introduction and discussion in a little more detail.”
I value this comment and again point out that the aforementioned second paragraph of the Introduction provides the detail sought by the reviewer.
Round 2
Reviewer 1 Report
The author already has numerous publications as an expert in this area of research. The present study is also well performed and written.
If I give a suggestion, the addition of other experimental tools and methods can be very powerful and more persuasive to claim the conclusions for future studies.
Author Response
“The author already has numerous publications as an expert in this area of research. The present study is also well performed and written.”
“If I give a suggestion, the addition of other experimental tools and methods can be very powerful and more persuasive to claim the conclusions for future studies.”
I appreciate this comment when it comes to locating precisely wherein the venom molecules are modulated. However, when it comes to detecting the functional modifications of these enzymes, the use of the thrombelastograph is the most useful mechanistically. I have added the following sentence in the Discussion, highlighted in purple for the reviewer to see:
“Lastly, the coupling of thrombelastographic methods with molecular analysis of purified, Ru-based molecule modified enzymes with tools such as mass spectroscopy will provide greater insight in future studies beyond the scope of the present work.”